# Nobiletin Inhibits Hypoxia-Induced Placental Damage via Modulating P53 Signaling Pathway

**DOI:** 10.3390/nu14112332

**Published:** 2022-06-01

**Authors:** Meng-Ling Zhang, Qian Yang, Yan-Di Zhu, Ya-Di Zhang, Rui Zhang, Jian Liu, Xiao-Yan Zhao, Qin-Yu Dang, Dong-Xu Huang, Ming-Yuan Zhang, Yu-Chen Wei, Zhuo Hu, Xia-Xia Cai, Li-Fang Gao, Yang Shan, Huan-Ling Yu

**Affiliations:** 1School of Public Health, Beijing Key Laboratory of Environmental Toxicology, Capital Medical University, Beijing 100069, China; 306855@ccmu.edu.cn (M.-L.Z.); syyq0903@163.com (Q.Y.); zhuyandi209@126.com (Y.-D.Z.); xgzzyd@126.com (Y.-D.Z.); zxyliunianwuzhuang@163.com (X.-Y.Z.); dang_qinyu@163.com (Q.-Y.D.); 17801061801@163.com (D.-X.H.); zmy598426400@163.com (M.-Y.Z.); weiyuchen_98@126.com (Y.-C.W.); huzhuo99@163.com (Z.H.); caixx1988@ccmu.edu.cn (X.-X.C.); lifanggao@ccmu.edu.cn (L.-F.G.); 2Hunan Agricultural Product Processing Institute, Hunan Academy of Agricultural Sciences, Hunan Provincial Key Laboratory for Fruits and Vegetables Storage Processing and Quality Safety, Changsha 410082, China; chinahunanliujian@163.com; 3Hunan Provincial Key Laboratory for Fruits and Vegetables Storage Processing and Quality Safety, Changsha 410082, China; 4Hunan Province International Joint Lab on Fruits & Vegetables Processing, Quality and Safety, Changsha 410082, China; 5Longping Branch Graduate School, Hunan University, Changsha 410082, China; 6School of Medical Humanity, Peking University, Beijing 100191, China; zhangrui2500@126.com

**Keywords:** Nobiletin, placenta, hypoxia, RUPP, P53 signaling pathway

## Abstract

In this study, we aimed to evaluate the effect of Nobiletin (NOB) on the placenta of Sprague–Dawley (SD) rats that had undergone reduced uterine perfusion pressure (RUPP) surgery and to evaluate the safety of NOB intervention during pregnancy. The results showed that NOB alleviated placental hypoxia, attenuated placental cell apoptosis, and inhibited placental damage in RUPP rats. No side effect of NOB intervention during pregnancy was observed. BeWo cell lines with P53 knockdown were then constructed using lentiviral transfection, and the P53 signaling pathway was found to be essential for NOB to reduce hypoxia-induced apoptosis of the BeWo cell lines. In summary, NOB attenuated hypoxia-induced placental damage by regulating the P53 signaling pathway, and those findings may contribute some insights into the role of NOB in placental development and the prevention of placental-related diseases.

## 1. Introduction

Placental insufficiency refers to a medical condition in which oxygen and nutrients are not sufficiently transferred to the fetus via the placenta during pregnancy, affecting about 10% of all pregnancies [1]. The underlying cause of placental insufficiency is typically a result of disturbances to the perfusion or blood supply of the placenta [2]. Any restrictions in the placental blood flow can lead to hypoxia [3]. Hypoxia promotes trophoblast cell apoptosis and triggers placental dysfunction by regulating the expression of the hypoxia-inducible factor 1α (HIF1α) and its target genes [1]. As a result, a series of placental diseases such as preeclampsia, intrauterine growth restriction, and placental abruption could emerge and wreak havoc on the life-long health of the offspring [4,5]. The mechanism of placental insufficiency is not fully understood, and it is generally believed that placental insufficiency is related to many factors such as oxidative stress, inflammation, and immunity. As yet, even in very early pregnancy, placental insufficiency always necessitates the termination of pregnancy [6]. Premature labor due to placental insufficiency always results in high neonatal morbidity and mortality [7].

Citrus-pomace-originated Nobiletin (5,6,7,8,3′,4′-hexamethoxyflavonoids, NOB) (Figure 1A) is an easily absorbable food-derived phytochemical with a higher lipophilicity and permeability than hydroxyflavonoids because of the six methoxy groups [8]. NOB is well-tolerated and non-toxic to humans and has been demonstrated to hold pharmacological activities including anti-oxidative stress, anti-inflammation, anti-tumor, and cardiovascular protection [9]. Only a few cases in the literature have reported that NOB shows anti-inflammatory effects by inhibiting the expression and production of pro-inflammatory cytokines in the placenta of diabetic pregnant mice [10]. The effects of NOB on hypoxia-induced placental insufficiency remain unclear.

Growing evidence suggests that NOB is effective against external stimulation-related cell apoptosis. NOB alleviates concanavalin A-induced acute liver injury in mice by targeting c-Jun N-terminal kinase [11]. NOB treatment effectively attenuates brain neuronal cell apoptosis, alleviating cerebral infarction and neurological damage in the cerebral hypoxia-reperfusion of male Wistar rats [12]. NOB reduces cardiomyocyte apoptosis to relieve cardiac insufficiency in male C57BL/6 mice undergoing myocardial infarction surgery [13]. Because of the inherent low-oxygen tension of the placenta, NOB may not have the same effect on the placenta as other tissues/cells under hypoxia [14]. Preliminary studies have demonstrated the safety of NOB on human placental trophoblast cells [15] as well as its effectiveness against hypoxia-induced apoptosis [16]. It is rational to hypothesize that NOB can protect against placental insufficiency caused by hypoxia.

In the present study, we sought to use the pregnant rat model of reduced uterine perfusion pressure (RUPP) to investigate the effects of NOB on placental injury and to explore the role of P53 signaling.

## 2. Material and Methods

### 2.1. Chemicals and Reagents

Dimethyl sulfoxide (DMSO), hematoxylin, and eosin were purchased from Solarbio (Beijing, China). The Rat HIF1α ELISA kit, Rat citrate synthase (CS) ELISA kit, Rat P53 ELISA kit, Rat P21 ELISA kit, and Rat BCL2 ELISA kit were obtained from Lunchangshuo (Xiamen, China). The Lactate dehydrogenase (LDH) kit was obtained from Jiancheng (Nanjing, China). The CF488 TUNEL cell apoptosis detection kit, 4% paraformaldehyde, and proteinase K were purchased from Servicebio (Wuhan, China). The DAPI anti-fluorescence quenching mounting tablets were obtained from Yeasen (Shanghai, China). The Trizol was from Invitrogen (Carlsbad, CA, USA). The Revert aid first strand cDNA synthesis kit was purchased from Thermo (Waltham, MA, USA). The TransStart^®^ Tip Green qPCR SuperMix qPCR was from Trans (Beijing, China). The Wes^TM^ automatic Western blot quantitative analysis kit was purchased from Protein Simple (San Jose, CA, USA). The RIPA lysis solution and PMSF were from Beyotime (Shanghai, China). The Polybrene promoter and puromycin were obtained from Hanbio Technology (Shanghai, China). theF12 (Ham) medium was obtained from Boster Biological Technology (Pleasanton, CA, USA). Fetal bovine serum (FBS) and dulbecco’s phosphate-buffered saline (DPBS) were from Corning (New York, NY, USA). The CCK-8 kit was purchased from Multi Sciences (Hangzhou, China). The Penicillin-streptomycin, Trypsin-EDTA solution, Annexin V/PI cell apoptosis detection kit, cell cycle detection kit, and BCA protein detection kit were purchased from Keygen (Beijing, China). ECL luminescent fluid was purchased from GE Healthcare (Buckinghamshire, UK). Cobalt chloride hexahydrate was purchased from Aladdin Biochemical Technology (Shanghai, China). Triton X-100 was obtained from Bioss (Beijing, China).

NOB (purity > 95%) for animal experiments was obtained from Kanglu (Hunan, China), and NOB for cell experiments (HPLC ≥ 95%) was purchased from Aladdin (Shanghai, China). The animal experiments used normal saline containing 0.1% DMSO to dissolve NOB. The medium used in cell experiments was prepared by dissolving NOB in F12 (Ham) medium containing 0.1% DMSO.

For Western blot analysis, the following antibodies were used: anti-ACTB Ab (1/2000; CST, MA, USA), anti-cl-PARP Ab (1/1000; Abcom, Cambridge, UK); anti-MDM2 Ab (1/1000; Abcom, Cambridge, UK), anti-BAX Ab (1/1000; Abcom, Cambridge, UK), and horseradish peroxidase-conjugated goat anti-mouse or rabbit IgG secondary antibody (1/2000; Abcom, Cambridge, UK). For Wes^TM^ automatic Western blot quantitative analysis system, the following antibodies were used: anti-ACTB Ab (1/500; CST, MA, USA), anti-cl-PARP Ab (1/200; Abcom, Cambridge, UK), anti-MDM2 Ab (1/200 Abcom, Cambridge, UK), and anti-BAX Ab (1/200; Abcom, Cambridge, UK). Goat anti-mouse or rabbit IgG secondary antibody had been included in the Wes^TM^ automatic western blot quantitative analysis kit.

The lentivirus used for P53 gene knockdown in BeWo cells was synthesized by Hanbio Technology (Shanghai, China). The top strand was GATCCGCAACTTGAGGAAGTACCATTATATTTCAAGAGAATATAATGGTACTTCCTCAAGTTGCTTTTTTG, and the bottom strand was AATTCAAAAAAGCAACTTGAGGAAGTACCATTATATTCTCTTGAAATATAATGGTACTTCCTCAAGTTGCG.

### 2.2. Animals and Treatment

All animal protocols and procedures used in this study were approved by the ethics committee of Capital Medical University (AEEI-2020-198). The specific pathogen-free (SPF) male and female Sprague–Dawley (SD) rats (7–8 weeks old) were purchased from Weitong Lihua Laboratory Animal Technology Co., Ltd. (Beijing, China). The rats were individually housed in conventional polypropylene cages (48 cm long, 33 cm wide, and 21 cm high) equipped with stainless steel food funnels, and the floor was covered by sawdust as the bedding. The cages were replaced twice a week. The room was kept at 20 ± 2 °C, with 70% humidity, and a 12 h light–dark cycle. All rats were given ad libitum access to a commercial rodent breeding diet (Keao xieli, Beijing, China) and water.

After one week of adaptive feeding, the virgin female rats were time-mated to male rats. The gestational day (GD) 0.5 was designated by the presence of the seminal plug or sperm in the vagina of the female rats. All pregnant rats were healthy and in a good mood, and were then randomly divided into four groups. (1) In the control group, the rats received a daily intraperitoneal injection of 0.1% DMSO–saline solution (total volume 200 uL) from GD 0.5 to GD 17.5. (2) In the NOB group, the rats received a daily intraperitoneal injection of 50 mg/kg body weight NOB (in 0.1% DMSO–saline solution, total volume 200 uL) from GD 0.5 to GD 17.5. (3) In the RUPP group, the rats received a daily intraperitoneal injection of 0.1% DMSO-saline solution (total volume 200 uL) from GD 0.5 to GD 17.5. On GD 18.5, the rats were subjected to the RUPP surgery according to the methods reported in the literature with minor modifications [17]. In detail, the rats were anesthetized with isoflurane before isolating the abdominal aorta. We ligated the abdominal aorta above the iliac bifurcation to impede blood flow to the placenta. However, the placenta still has a compensatory blood flow through an adaptive increase in ovarian blood flow [18]. Therefore, the branches of the two ovarian arteries that supply the uterus were also ligated to prevent blood from flowing into the uterus through the ovarian arteries. (4) In the RUPP + NOB group, the rats received a daily intraperitoneal injection of 50 mg/kg body weight NOB (in 0.1% DMSO-saline solution, total volume 200 uL) from GD 0.5 to GD 17.5. On GD 18.5, the rats were weighed and subjected to RUPP for eight hours. After the procedure, the pregnant rats were anesthetized and sacrificed. The placenta, fetuses, and maternal tissues (brain, heart, liver, spleen, lung, and kidney) were separated, weighed, and photographed. One-half of the specimens were fixed with 4% paraformaldehyde at room temperature; and the other half were preserved in liquid nitrogen and then stored at −80 °C.

### 2.3. Reverse Transcription (RT)-and Real-Time Quantitative (q) PCR

The total RNA was extracted from the placentas and the cultured cells using Trizol reagent. The reverse transcription and real-time qPCR were conducted using a RevertAid First Strand cDNA Synthesis Kit and a TransStart^®^ Tip Green qPCR SuperMix qPCR Kit, respectively. The validated primers were purchased from Sangon Biotech (Beijing, China). The primers used for the qPCR are shown in Table 1. The parameters of PCR were 95 °C for 4 min, followed by 39 cycles for 10 s at 95 °C, and 45 s at 60 °C.

### 2.4. Hypoxia-Related Biomarkers

Tissues and cells were lysed in the RIPA lysis buffer containing 1% PMSF, and protein levels were quantified using the BCA protein assay kit. HIF1α of the placenta tissue was detected by the ELISA kit. The activities of LDH and CS were determined by the LDH and CS kit, respectively.

### 2.5. Hematoxylin-Eosin (H&E) Staining

The placental tissue fixed in 4% paraformaldehyde was dehydrated, paraffin-embedded, sectioned, deparaffinized, and stained with hematoxylin and eosin. Then, the slices were dehydrated and mounted with neutral gum. The dried slices were scanned by a pathological slice scanner (Pannoramic SCAN, Budapest, Hungary), and the data were collected. The area of the placenta, placental junction, and placental labyrinth was quantified using Pro Plus 6.0 software (Silver Spring, MD, USA).

### 2.6. TUNEL Staining

The severity of placental apoptosis was tested by a CF488 TUNEL cell apoptosis detection kit. The placental tissues fixed in 4% paraformaldehyde were dehydrated, paraffin-embedded, sectioned, and deparaffinized. The slices were then placed in a wet box and washed with DPBS three times. A PAP pen was used to draw a circle about 2–3 mm apart from the tissue following the outline of the tissue. The slices were kept wet throughout the experiment. Freshly prepared proteinase K (20 μg/mL, 100 μL) was added to the slices which were then incubated at 37 °C for 20 min. The slices were treated with 0.1% Triton X-100 containing 0.1% sodium citrate and incubated again at 37 °C for 20 min. The labeling was performed according to the protocol, and the scanned images by a pathological slice scanner were collected. The positive apoptotic cells were counted using Image Pro Plus 6.0 software.

### 2.7. Western Blot

Tissues and cells were lysed in the RIPA lysis buffer containing 1% PMSF, and protein levels were quantified using the BCA protein assay kit. 20 μg of total proteins were electrophoresed by 10% or 12% sodium dodecyl sulfate polyacrylamide gel, and the separated proteins were transferred to the nitrocellulose membrane. After blocking and washing, the membranes were incubated with the specific primary antibody overnight at 4 °C and then washed with TBST before incubation with the HRP-conjugated secondary antibody for 1 h. The membrane with ECL luminescent liquid added was exposed to the gel imaging system, and the grayscale of the protein band was calculated using Image J software (Bethesda, MD, USA). The ACTB was the internal reference protein in this experiment. The relative expression of the target protein was defined as the ratio of the grayscale of the target protein to the internal reference protein.

### 2.8. Cells, Lentiviral Transfection, and Treatment with NOB

The human placental choriocarcinoma BeWo cell line was purchased from Peking Union Medical College (Beijing, China). BeWo cells were seeded in a sterile culture flask at a density of 1 × 10^5^ cells/mL and incubated in an F12 (Ham) medium containing 15% FBS and 100 U/mL penicillin–streptomycin at 37 °C in a humidified incubator with 5% CO_2_. The cell culture media were changed every three days. When the cells covered 80–90% of the medium, they were digested with trypsin and passaged.

Stably transfected BeWo cell lines with a P53 gene knockdown were constructed using lentivirus, and BeWo cell lines with an empty vector were also constructed. The expression efficiency of the virus with the GFP reporter gene was detected using a fluorescence microscope. Stably transfected cell lines with the puromycin resistance gene virus were selected using the complete medium containing 2 mg/mL puromycin. RT-qPCR was used to detect the expression of the target gene to ensure the successful knockdown of the P53 gene in BeWo cell lines.

BeWo cells with a P53 gene knockdown and an empty vector were seeded at a density of 1 × 10^5^ cells/mL and cultured for 72 h. After removing the medium, they were randomly divided into five groups: (1) BeWo cells with an empty vector were cultured in the serum-free F12 (Ham) medium containing only 0.1% DMSO for 12 h. (2) BeWo cells with an empty vector were cultured in the serum-free F12 (Ham) medium containing 500 μM CoCl_2_ and 0.1% DMSO for 12 h. (3) BeWo cells with an empty vector were cultured in the serum-free F12 (Ham) medium containing 500 μM CoCl_2_, 100 μM NOB, and 0.1% DMSO for 12 h. (4) BeWo cells with a P53 gene knockdown were cultured in the serum-free F12 (Ham) medium containing 500 μM CoCl_2_ and 0.1% DMSO for 12 h. (5) BeWo cells with a P53 gene knockdown were cultured in the serum-free F12 (Ham) medium containing 500 μM CoCl_2_, 100 μM NOB, and 0.1% DMSO for 12 h. After the intervention, the cells were collected for subsequent tests.

### 2.9. CCK-8 Assay

Cell viability was detected using a CCK-8 kit. After the intervention (see Section 2.8), the cells were removed from the medium, washed with DPBS three times, and treated with the CCK-8 solution of 10 μL/well. After incubating at 37 °C for 4 h, the absorbance at 450 nm was measured using a microplate reader (Biotek, Winooski, VT, USA).

### 2.10. Cell Morphology

After the intervention (see Section 2.8), cell morphology was monitored using an optical microscope.

### 2.11. Cell Cycle Distribution

Cell cycle distribution was measured by a cell cycle kit. The BeWo cells were seeded into a 6-well plate and incubated for 72 h. After the intervention (see Section 2.8), the cells were washed, trypsinized, fixed with 70% ethanol, and stored at 4 °C overnight. The cells were then washed with ice-cold DPBS and incubated with the PI staining buffer in a dark box for 30 min at room temperature. The cells in the subG1, G1, S, and G2/M phases were measured using a flow cytometer (ACEA, San Diego, CA, USA).

### 2.12. Wes^TM^ Automatic Western Blot Quantitative Analysis

BeWo cells were lysed with RIPA lysis buffer containing 1% PMSF, and the total proteins were quantified using the BCA kit. The levels of the specific protein were detected using a Wes^TM^ automatic Western blot quantitative analysis system. 2.25 μg of the total proteins were loaded, and the molecular weight of the proteins separated by the capillary ranged from 12 to 230 kDa. ACTB was the internal reference protein in the experiment. The relative expression of the target protein was defined as the ratio of the fluorescence value of the target protein to that of the internal reference protein.

### 2.13. Statistical Analysis

All experiments were independently conducted three or more times, and the data were expressed as mean ± standard error (Mean ± SEM). The one-way ANOVA was performed using Statistical Analysis System software (SAS Institute Inc., Cary, NC, USA). The means of different groups were compared using a Duncan multiple range test (*p* < 0.05). All pictures were generated using Graph Pad Prism software (Version 6.00, Graph pad Software Inc., San Diego, CA, USA) and typeset in Adobe Illustrator software (Adobe illustrator cc 2017, San Jose, AR, USA). All forms were produced using Microsoft Office Word (Redmond, WA, USA).

## 3. Results

### 3.1. NOB Inhibited Placental Damage in RUPP Rats

#### 3.1.1. NOB Alleviated Placental Hypoxia in RUPP Rats

To investigate the effect of NOB during pregnancy on RUPP rats and fetuses, the weight and organ coefficient of the female rats were examined. The markers of fetal growth restriction, such as litter size, fetal weight, and placental efficiency, were compared. The liver and spleen coefficient of the female rats was lower in the RUPP and NOB + RUPP groups than in the control groups (Figure 1B). The weight and other organ coefficients (brain, heart, lung, and kidney) of the female rats had no significant differences between all experimental groups, nor did the litter size, fetal weight, and placental efficiencies (Figure 1B–D).

We also examined the placental hypoxia biomarkers, such as HIF1α levels, LDH, and CS activities. HIF1α levels, LDH, and CS activities in the placental tissues were increased in the RUPP groups in comparison to the control groups (Figure 2A). In the RUPP + NOB groups, the HIF1α levels, as well as the LDH and CS activities of the placenta, were decreased compared with the RUPP groups, suggesting that NOB inhibited placental hypoxia in the RUPP rats. The HIF1α levels of the placenta of the NOB groups were similar to the control groups, and only a slight decrease in LDH and CS activities in the placenta of the NOB groups was observed.

#### 3.1.2. NOB Alleviated Placental Pathological Injury in RUPP Rats

To clarify the protective effect of NOB on hypoxia-induced placental injury, the pathological changes in the placenta were measured using H&E staining. The rats that had undergone RUPP surgery had fewer blood cells in the placental labyrinth area than the control groups, indicating that placental nutrients and gas exchange after placental hypoxia were impaired, but an alleviation was noticed in the RUPP + NOB groups (Figure 2B). No differences in placental morphology were detected between the NOB groups and the control groups. There were no significant differences in placental thickness, placental area, junction area, and labyrinth area among all the test groups. In the RUPP groups, the ratio of the placental junction area to the labyrinth area was lower than that of the control groups, whereas that trend was reversed after NOB pretreatment.

#### 3.1.3. NOB Alleviated Placental Apoptosis in RUPP Rats

Apoptosis, an important factor in placental damage, is measured by TUNEL staining and cl-PARP levels. After RUPP surgery, there was an appreciable increase in the TUNEL-positive cell number and cl-PARP protein level in the placenta than in the control groups, but no such increase was observed after NOB pretreatment (Figure 2C,D). No significant difference in the proportion of TUNEL-positive cells and the level of cl-PARP protein in the placenta was found between the NOB groups and the control groups.

#### 3.1.4. NOB Inhibited the Activation of Placental P53 in RUPP Rats

P53, often referred to as the “guardian of the genome”, is a key tumor suppressor that monitors hypoxic stress signals by controlling specific transcription targets for cell cycle arrest and apoptosis [19]. The placental mRNA and protein levels of P53, MDM2, and P21 in the RUPP groups were higher than in the control groups, and the increase in the RUPP groups was also more pronounced than in the RUPP + NOB groups (Figure 3A,B). This activated P53 induced apoptosis by regulating the apoptotic pathway in mitochondria and regulating the activities of anti-apoptotic proteins and pro-apoptotic proteins of BCL2 family proteins [20]. Generally, the ratio of BCL2/BAX indicates the sensitivity of cells to apoptosis [21]. The ratio of BCL2/BAX mRNA and protein of placental cells in the RUPP groups were significantly lower than those of the control groups. After pretreatment with NOB during pregnancy, the ratios of BCL2/BAX mRNA and protein in placental cells of RUPP + NOB rats were significantly higher than those of RUPP rats. The above results suggested that NOB alleviated the apoptosis of rat placental cells induced by hypoxia.

### 3.2. The Anti-Apoptotic Effect of NOB Partly Depended on the Inhibition of P53

#### 3.2.1. Effects of NOB on Cell Proliferation of BeWo Cell Line with p53 Knockdown

To further investigate the possible role of P53 for NOB to attenuate hypoxia-induced placental cell apoptosis, BeWo cells with a P53 gene knockdown were used in vitro experiments. The cell viability and cell number had a remarkable reduction in the hypoxic cells compared with the control. The treatment of NOB increased the cell viability of hypoxic cells (Figure 4A,B). The cell viability of normal cells with a P53 gene knockdown decreased significantly, but interestingly, NOB treatment did not change the cell viability in this case.

The results of flow cytometry demonstrated the effect of NOB on the cell cycle distribution of BeWo cell lines. The hypoxia groups had more cells arrested in the G1 phase than the control groups, but the number of G1 cells had dropped in the groups with NOB intervention (Figure 4C). There were significantly fewer G1-phase-arrested cells in the P53-knockdown groups than in the hypoxia groups. However, there was no difference seen in the proportion of G1-phase-arrested cells between the two P53-knockdown groups with or without NOB intervention.

#### 3.2.2. Effects of NOB on Cell Apoptosis of BeWo Cell Line with p53 Knockdown

The severity of cell hypoxia and apoptosis was characterized by the HIF1α level and cl-PARP level, respectively. NOB limited the increase in the HIF1α level and cl-PARP level of BeWo cells induced by hypoxia (Figure 4D,E). In the P53-knockdown groups, cell hypoxia was significantly inhibited as the down-regulation of the HIF1α level compared with the hypoxia groups. After NOB intervention, the HIF1α protein level of P53 knockdown cells further declined, while the cl-PARP level was unchanged.

#### 3.2.3. Effects of NOB on P53 Signaling Pathway of BeWo Cell Line with p53 Knockdown

The mRNA and proteins concerning the P53 signaling pathway in BeWo cells were detected using an RT-qPCR and Wes. When compared with the control group, hypoxia caused an increase in the level of the mRNA and protein of P53, MDM2, and P21. After NOB intervention, the level of the mRNA and protein of P53 and P21 was lower than in the hypoxia group without NOB treatment, but the change related to MDM2 was not significant (Figure 5A,B). In the P53-knockdown group, the level of the above mRNA and proteins of P53, MDM2, and P21 decreased remarkably compared with the hypoxia group, and it was not changed by NOB treatment. These results suggested that the P53 signaling pathway played an important role in NOB’s anti-apoptotic effect under hypoxia conditions.

## 4. Discussion

This study showed the impact of NOB intervention on placental insufficiency in the RUPP pregnant rat model, and also assessed the safety of NOB intervention during pregnancy. Supplementation with NOB during pregnancy reduced the P53 level and cell apoptosis and alleviated placental insufficiency in RUPP rats. NOB intervention during pregnancy had no side effects on the placenta in comparison with the control groups. BeWo cell lines with a P53 knockdown showed that the P53 signaling pathway was one of the crucial factors for alleviating hypoxia-induced BeWo cell apoptosis.

Out of the placental hypoxia models available, the pregnant rat with RUPP operation is one of the most reproducible and well-characterized models [22]. In this model, the perfusion to the utero-placental circulation of pregnant rats is partially blocked, thereby simulating the decrease in perfusion due to impaired spiral artery remodeling [23]. RUPP procedure impairs placental blood flow, triggering enhanced HIF1α expression and placental hypoxia [24]. RUPP can also damage the structure of the placental junction and labyrinth areas, causing placental insufficiency [25]. Studies have reported that placenta defects in RUPP rats caused a series of maternal and fetal pathological changes, including increased maternal blood pressure, proteinuria, and fetal growth restriction [6,23]. In the present study, the RUPP model was used to investigate the effect of NOB intervention during pregnancy on mediating placental hypoxic injury during the chronic reduction of uterine perfusion. The results showed that the RUPP procedure in rats caused hypoxia in the placenta, reduced the number of blood cells in the labyrinthine area of the placenta, decreased the ratio of the placental junction area to the labyrinth area, and caused obvious placental apoptosis.

Clinically, the diseases associated with placental defects caused by hypoxia can be prevented by aspirin [26], metformin [27], vitamin C [28], vitamin E [29], and melatonin [30], which are limited by potential high-dose toxicity or only a short-term effect. Therefore, it is necessary to explore more efficient and safe phytochemicals to prevent placental insufficiency. A meta-analysis reported that heparin improved the neonatal outcomes of fetuses with placental insufficiency [31]. However, the article only included randomized controlled trials, and there were insufficient quality data and evidence to recommend heparin for the treatment. Maternal choline supplementation has been proved to modulate placental apoptosis due to placental insufficiency in a mice model. However, many pregnant women do not currently meet the recommended intake of choline, and the effects of increasing maternal choline intake on placental function in humans are still not fully understood [32]. In our study, NOB intervention during pregnancy (50 mg/kg NOB per day) caused a decrease in the number of TUNEL-positive cells and the protein level of cl-PARP in the placenta of rats undergoing RUPP, and alleviated the pathological damage of the placenta. In the United States, NOB has been approved as a nutritional supplement and is claimed to improve cognitive and metabolic function, promote cardiovascular function, and regulate circadian rhythm. NOB is expected to become a healthy food or new food ingredient approved in China. These evidences suggested that NOB is a potential phytochemical for the prevention and management of placental dysfunction.

As an important tumor suppressor gene, P53 forms a tetrameric transcription factor that directly regulates about 500 target genes, thereby controlling cell processes including DNA breaks, apoptosis, and cell cycle caused by hypoxia [33]. Under mild hypoxia, P53 protein levels are decreased to protect cells from apoptosis and promote cell viability. In contrast, under severe hypoxia, P53 protein levels are stabilized, resulting in the reduced HIF1α transcriptional activity and apoptosis [34]. A preliminary study found that NOB inhibited the hypoxia-induced apoptosis of JEG-3 and BeWo trophoblast cells, and the severity of apoptosis was positively correlated to the level of P53 [16]. Molecular docking and molecular dynamics simulation found that NOB bonds to the P53 protein by reducing free energy and interacts with Leu145, Thr150, Pro151, Thr155, Val157, Cys220, Pro222, Pro223, and Thr230 in a flexible loop region of the P53 protein [16]. The results of UV spectroscopy, fluorescence spectroscopy, and circular dichroism further proved that NOB could spontaneously bind to the P53 protein, enter the hydrophobic pocket of the P53 protein, cause the static quenching of the P53 protein, change the protein conformation, and finally affect the function and activity of the P53 protein [16]. The above findings suggested that the P53 pathway may be one of the mechanisms by which NOB exerts anti-trophoblast apoptosis. To test this hypothesis, the in vitro experiments tested BeWo cell lines with the P53 gene knockdown. The results showed that NOB intervention alleviated hypoxia-induced apoptosis and cell cycle arrest in the G1 phase of BeWo cell lines; but in the cell line with P53 gene knockdown, NOB was less effective in alleviating apoptosis and cell cycle arrest. Those findings indicated that the P53 signaling pathway was critical for NOB’s anti-apoptosis effect. A schematic diagram of the probable molecular mechanism of NOB’s inhibiting effect on RUPP-induced placental damage was presented in Figure 6.

## 5. Conclusions

The present study demonstrated that NOB supplementation during pregnancy inhibited placental apoptosis and attenuated placental damage in an SD rat model of RUPP. No side effects were found in normal rats taking NOB during pregnancy. In this process, NOB could inhibit the P53 signaling pathway, resulting in the inhibition of the placental apoptosis and damage caused by RUPP. Therefore, the research provided a new approach to the prevention and clinical treatment of some placental hypoxia-related diseases, and contributed to the development and utilization of natural plant compounds.

## Figures and Tables

**Figure 1 nutrients-14-02332-f001:**
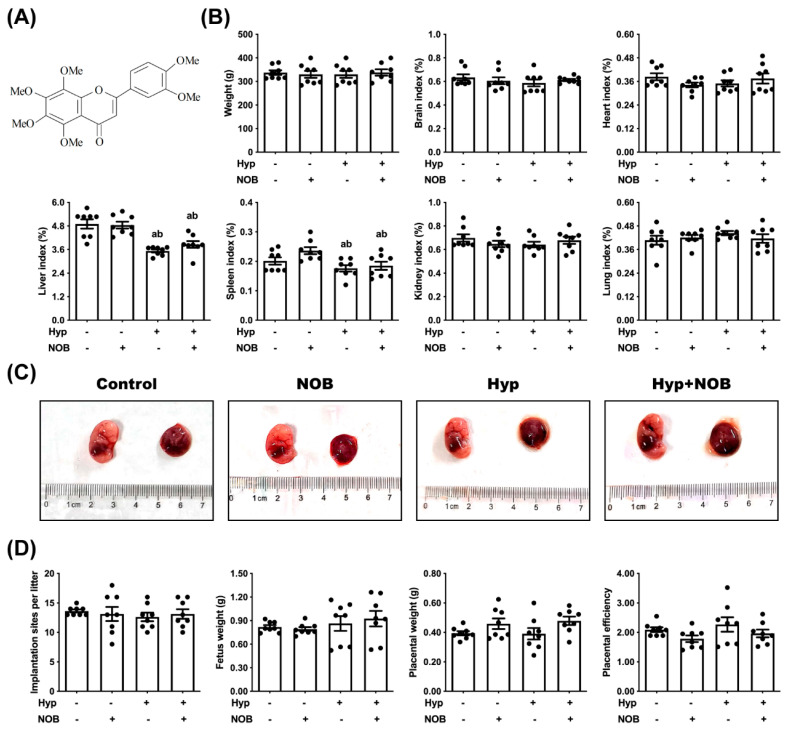
The influence of NOB intervention during pregnancy on the weight of the maternal rats, placenta, and fetus in RUPP-induced SD rats. (**A**) Chemical structure of Nobiletin (5,6,7,8,3,4-hexamethoxyflavone, NOB). (**B**) Body weight and organ coefficient of the pregnant rats. (**C**) Representative pictures of the placenta and fetus. (**D**) The implantation sites per litter, placental weight, fetus weight, and placental efficiency (fetal/placental weight ratio). Data are expressed as the mean ± standard error (Mean ± SEM). a: comparing with the control groups, *p* < 0.05; b: comparing with the NOB groups, *p* < 0.05.

**Figure 2 nutrients-14-02332-f002:**
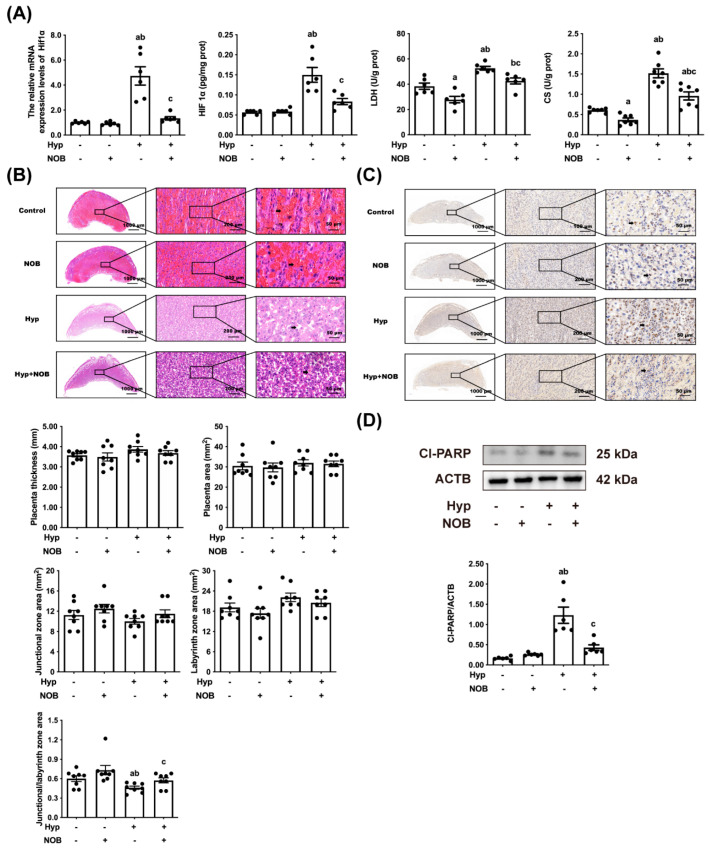
NOB attenuated hypoxia and apoptosis of rat placental cells caused by RUPP. (**A**) The HIF1α mRNA and protein levels of the placenta were tested using an RT-qPCR and a rat HIF1α ELISA kit, respectively. The LDH activity and CS activity of the placenta were estimated using an LDH kit and a rat CS ELISA kit, respectively. (**B**) Representative pictures of placental histology of H&E staining in a similar place in the cross-cut dimension, as well as the placental thickness, placental area, junctional zone area, labyrinth zone area, and the ratio of the placental junction area to the labyrinth area. (**C**) Representative pictures of DNA fragmentation in the placenta of pregnant rats by TUNEL staining in a similar place in the cross-cut dimension. (**D**) The protein expression of cl-PARP in the placental lysate was detected by Western blot and the relative expression levels were quantified. ACTB was used as an internal control, and the relative protein expression was the ratio of the target protein to ACTB. Data are expressed as the mean ± standard error (Mean ± SEM). a: comparing with the control groups, *p* < 0.05; b: comparing with the NOB groups, *p* < 0.05; c: comparing with the hypoxia groups, *p* < 0.05.

**Figure 3 nutrients-14-02332-f003:**
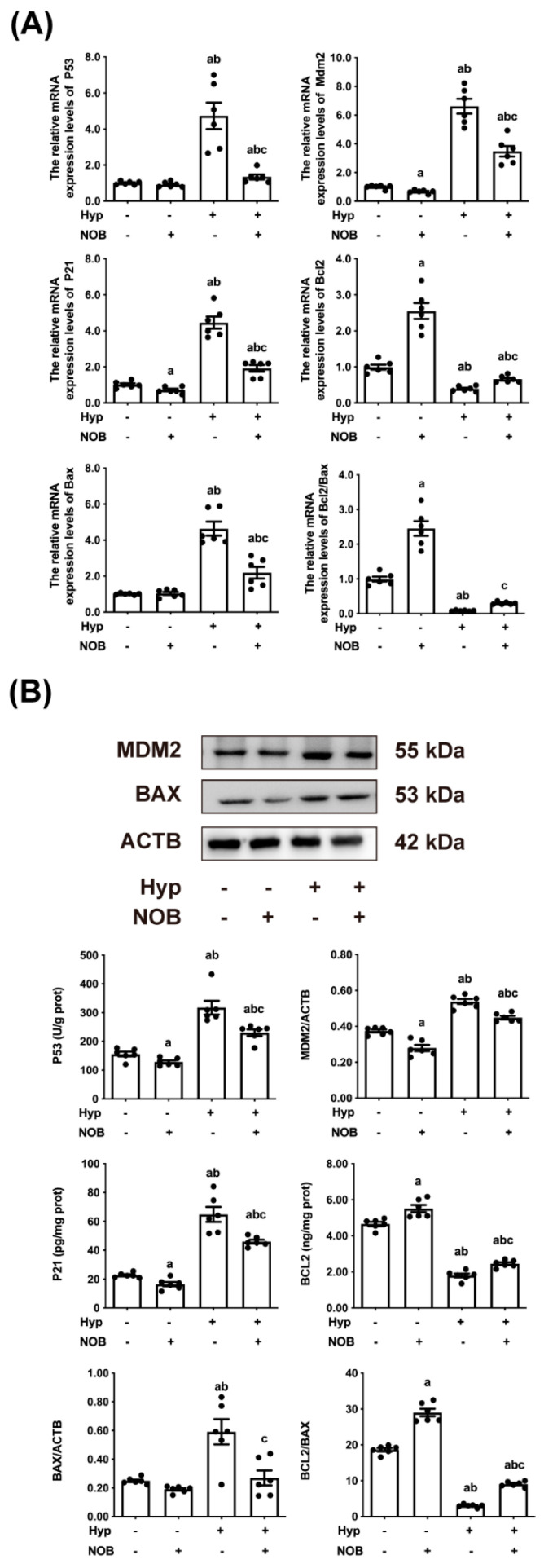
NOB modulated placental P53 pathway and mitochondrial apoptosis pathway in RUPP rats. (**A**) The mRNA expressions of P53, Mdm2, P21, Bcl2, and Bax were estimated using an RT-qPCR. The mRNA ratio of Bcl2/Bax was calculated. ACTB was used as an internal control. (**B**) The protein expressions of P53, P21, and BCL2 were detected using the rat P53 ELISA kit, rat P21 ELISA kit, and rat BCL2 ELISA kit, respectively. The MDM2 and BAX protein levels were estimated using a Western blot. ACTB was used as an internal control, and the relative protein expression was the ratio of the target protein to ACTB. Data are expressed as the mean ± standard error (Mean ± SEM). a: comparing with the control groups, *p* < 0.05; b: comparing with the NOB groups, *p* < 0.05; c: comparing with the hypoxia groups, *p* < 0.05.

**Figure 4 nutrients-14-02332-f004:**
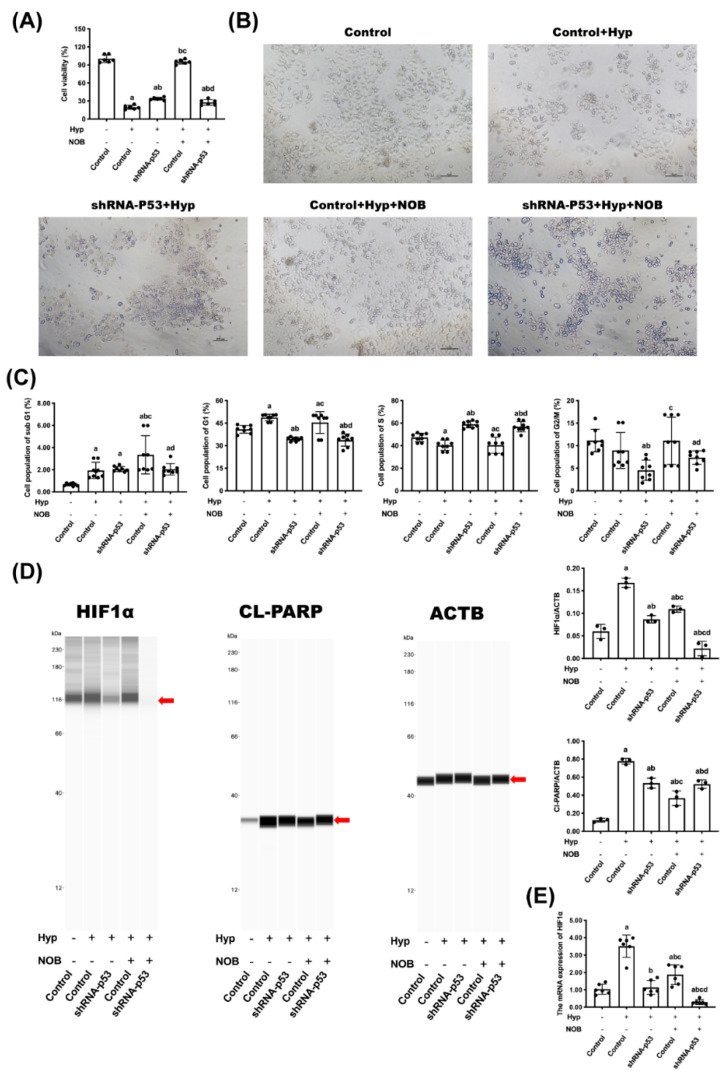
NOB did not change the proliferation and apoptosis of BeWo cell lines after P53 knockdown. (**A**) The cell activity was measured using a CCK-8 assay. (**B**) The morphologies of cells were recorded using a microscope. (**C**) The cell cycle distribution of cells was detected using flow cytometry with PI staining. (**D**) The protein expressions of HIF 1α and cl-PARP in cell lysate were detected using Wes, and the relative expression levels were quantified. ACTB was used as an internal control, and the relative protein expression was the ratio of the target protein to ACTB. (**E**) The mRNA expressions of HIF 1α were estimated using an RT-qPCR. ACTB was used as an internal control. Data are expressed as the mean ± standard error (Mean ± SEM). a: comparing with the control groups, *p* < 0.05; b: comparing with the hypoxia groups, *p* < 0.05; c: comparing with the hypoxia+shRNA-P53 groups, *p* < 0.05; d: comparing with the hypoxia+NOB groups, *p* < 0.05.

**Figure 5 nutrients-14-02332-f005:**
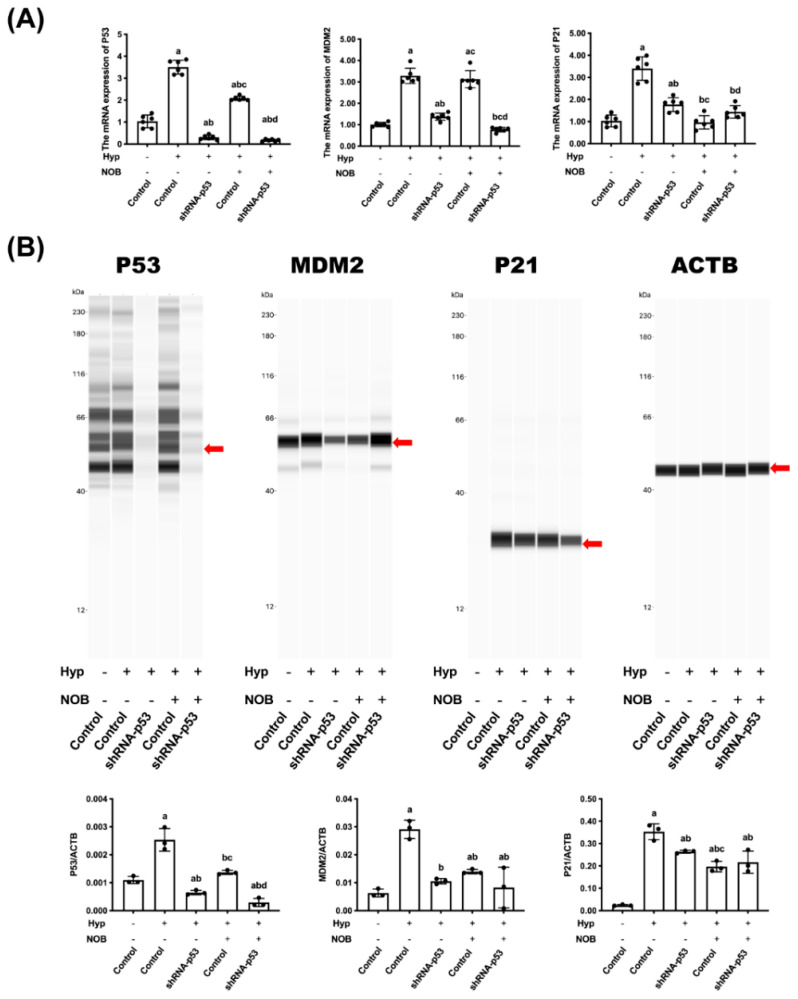
P53 signaling pathway played a critical role in alleviating the hypoxia-induced apoptosis of BeWo cells. (**A**) The mRNA expressions of P53, MDM2, and P21 were estimated by RT-qPCR. ACTB was used as an internal control. (**B**) The protein expressions of P53, MDM2, and P21 were detected using Wes. ACTB was used as an internal control, and the relative protein expression was the ratio of the target protein to ACTB. Data are expressed as the mean ± standard error (Mean ± SEM). a: comparing with the control groups, *p* < 0.05; b: comparing with the hypoxia groups, *p* < 0.05; c: comparing with the hypoxia+shRNA-P53 groups, *p* <0.05; d: comparing with the hypoxia + NOB groups, *p* < 0.05.

**Figure 6 nutrients-14-02332-f006:**
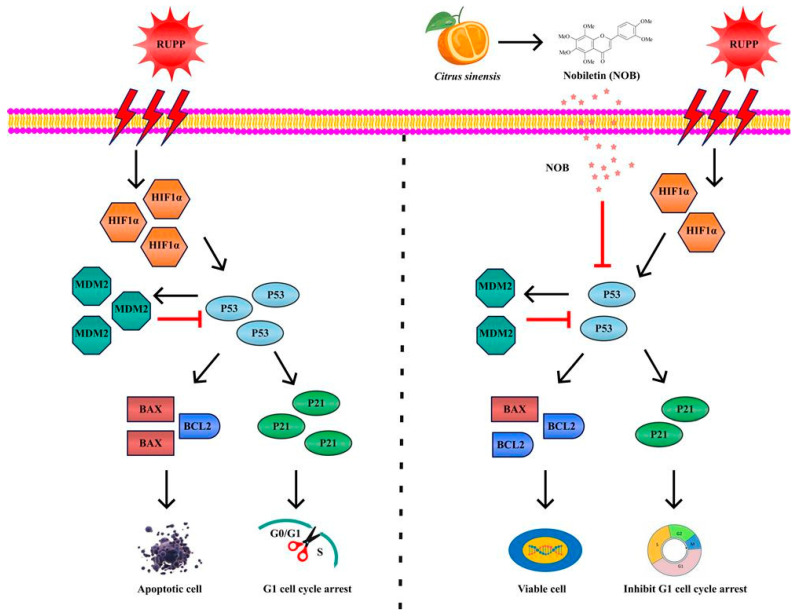
Putative mechanism of NOB’s inhibiting effect on RUPP-induced placental damage. NOB reduced the level of P53 in the placenta of RUPP rats, thereby inhibiting the apoptosis and relieving G1 cell cycle arrest.

**Table 1 nutrients-14-02332-t001:** Nucleotide sequences of the forward and reverse primers for RT-qPCR.

Gene	Species	Forward Primer	Reverse Primer
*Hif1α*	Rat	CGGCGAGAACGAGAAGAAAAATAGG	GACTCTTTGCTTCGCCGAGA
*P53*	Rat	CCCCTGAAGACTGGATAACTGT	TTAGGTGACCCTGTCGCTGT
*Mdm2*	Rat	CGAGCGAAATGGTCTCTCAAG	TGCAGACCGCTGCTACT
*P21*	Rat	AACAGGCTCAGGAGTTAGCA	CATCGTCAACACCCTGTCTT
*Bcl2*	Rat	AGTACCTGAACCGGCATCTG	TATAGTTCCACAAAGGCATCCCAG
*Bax*	Rat	ACCAAGAAGCTGAGCGAGTG	TCCACATCAGCAATCATCCTCT
*Actb*	Rat	ACCCGCGAGTACAACCTTCTT	TATCGTCATCCATGGCGAACTGG
*HIF1α*	Human	GGCGCGAACGACAAGAAAAAG	CCTTATCAAGATGCGAACTCACA
*P53*	Human	AAAAGTCTAGAGCCACCGTCC	AGTCTGGCCAATCCAGGGAAG
*MDM2*	Human	TCTCCTGCCTCAGCCTTCCAAG	GCCAGGTGCCTCACATCTGTAATC
*P21*	Human	CCTGTCACTGTCTTGTACCCT	GCGTTTGGAGTGGTAGAAATCT
*ACTB*	Human	TCCTTCCGCAGCTATTTATGAT	CACAGTATAGGATGGTCTGGAC

## Data Availability

The data are available from the corresponding author upon reasonable request.

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
