# Peer review of "Nobiletin Inhibits Hypoxia-Induced Placental Damage via Modulating P53 Signaling Pathway"

_nutrients, 2022, doi:10.3390/nu14112332_

Round 1
Reviewer 1 Report
In the study "Nobiletin reduced hypoxia-induced placental insufficiency via modulating P53 signaling pathway and global DNA methylation", Meng-ling Zhang and co-workers report the effect of
Citrus-pomace-originated nobiletin (NOB which is not defined as abbreviation in the abstract) on placenta hypoxia. Unfortunately is very difficult to review this manuscript in the current form, mainly because the way it is crafted. Several grammatical mistakes prevent understanding of the real authors' meaning. Moreover the result section should be re-written, with each paragraph starting with a brief description of what has been done, and why. For example the first paragraph starts in this way "Compared with the control group, RUPP and NOB+RUPP treatment decreased the liver and spleen index of pregnant rats (Fig.1B). There were no significant differences in weight (pregnant rats, fetuses and placenta), organ index (brain, heart, lung and kidney), litter size and placental efficiency among all experimental groups (Fig.1B-D)".
So the reader understand there were groups and treatments involved, but not much more. I would recommend following the pattern "In order to investigate XXX, we decided use YY. To this end, OOOO were ether treated with AAAA or left untreated. BBBB days post treatment Samples were collected and subjected to CCCC analysis. Our results show that DDDD, (Figure EEEE).
Overall, a profound re-structuring and re-writing of the manuscript is recommended before resubmission.
Author Response
Responses to reviewer A comments
Question one:
Unfortunately is very difficult to review this manuscript in the current form, mainly because the way it is crafted. Several grammatical mistakes prevent understanding of the real authors' meaning.
Answer:
Thank you for your suggestion, and we have revised the manuscript.
The major changes were marked in red.
Question two:
Moreover the result section should be re-written, with each paragraph starting with a brief description of what has been done, and why. For example the first paragraph starts in this way "Compared with the control group, RUPP and NOB+RUPP treatment decreased the liver and spleen index of pregnant rats (Fig.1B). There were no significant differences in weight (pregnant rats, fetuses and placenta), organ index (brain, heart, lung and kidney), litter size and placental efficiency among all experimental groups (Fig.1B-D)". So the reader understand there were groups and treatments involved, but not much more. I would recommend following the pattern "In order to investigate XXX, we decided use YY. To this end, OOOO were ether treated with AAAA or left untreated. BBBB days post treatment Samples were collected and subjected to CCCC analysis. Our results show that DDDD, (Figure EEEE). Overall, a profound re-structuring and re-writing of the manuscript is recommended before resubmission.
Answer:
Before revision:
Compared with the control group, RUPP and NOB+RUPP treatment decreased the liver and spleen index of pregnant rats (Fig.1B). There were no significant differences in weight (pregnant rats, fetuses and placenta), organ index (brain, heart, lung and kidney), litter size and placental efficiency among all experimental groups (Fig.1B-D).
After revision:
To investigate the effect of NOB during pregnancy on RUPP rats and fetuses, the weight and organ coefficient of the female rats were examined. The markers of fetal growth restriction, such as litter size, fetal weight, and placental efficiency were compared. The liver and spleen coefficient of the female rats was lower in the RUPP and NOB+RUPP groups than the control groups (Fig. 1B). No difference was observed in weight and other organ coefficients (brain, heart, lung and kidney) of the female rats litter size, fetal weight, and placental efficiency between all experimental groups (Fig. 1B-D).
The major changes were marked in red.
Reviewer 2 Report
The Authors evaluated the effect of NOB on the placenta of Sprague-Dawley rats that had undergone reduced uterine perfusion pressure (RUPP) surgery, and subsequently evaluated the safety of NOB intervention during pregnancy.
The paper is overall quite well written, however there some major concerns:
- the RUPP acronym is explained only in the abstract. The procedure should be described in the Materials&Methods section and the cited reference (#18) is quite outdated.
- Fig.1 should be positioned later in the text: in the present position is confusing, because it refers to Results.
- Some sentences are awkward, i.e. "As shown in Fig. 2A, the increase in placenta of HIF1α levels that was observed in RUPP groups as compared with control groups (page 7); or "This study provided a new insight into the prevention and treatment of placenta-related diseases by phytochemicals." (page 3) written in the introduction, while it sounds as a results/discussion argument.
- The legends of figures #1,2 and 3 are identical in the initial part. This is highly redundant and useless, as the procedures are explained in the Materials&Methods section.
- The conclusions sounds a bit reckless. Placental insufficiency is an umbrella term which encompass many different conditions (immunological such as LES, or anatomical, such as uterine malformations, or related to placental implantation site, such as placenta previa), any of which can lead to abruptio placentae, IUGR, fetal demise, stillbirth etc. : reduced blood flow/pressure from spiral uterine arteries (mimicked by RUPP procedure) is not the only mechanism by which placental hypoxia takes place in humans. Maternal vascular malperfusion (the histologic pattern of placental hypoxia) may show also signs of acute atherosis, fibrinoid necrosis, chronic perivasculitis, trophoblasts persistence in placental implantation bed etc. The Authors should avoid to draw such relations in the regards of suggesting that the daily intake of citrus juice could help to prevent placental insufficiency in pregnant women. with such easiness, and leave the good results they achieved in the experimental area.
- Many references are outdated and should be substituted with more recent ones.
- English revision is recommended.
Author Response
Responses to reviewer B comments
Question one:
The RUPP acronym is explained only in the abstract. The procedure should be described in the Materials & Methods section and the cited reference (#18) is quite outdated.
Answer:
Before revision:
On the GD 18.5, the rats were weighed and then performed with RUPP procedure for eight hours according to the report [18]. The steps were as follows: After anesthesia, the rats were incised in the midline of the abdomen, and the inferior abdominal aorta was separated and the aorta above the iliac branches was ligated. Compensatory blood flow to the placenta occurs through the adaptive increase of ovarian blood flow [19]. To prevent blood from flowing into the uterus through the ovarian artery, the branches of the two ovarian arteries that supply the uterus were also ligated.
After revision:
On the GD 18.5, Rats were subjected to RUPP surgery according to the methods reported in the literature with minor modifications (Williamson et al., 2020). In brief, under isoflurane anesthesia, RUPP inhibited the blood flow to the uteroplacental unit by ligating the abdominal aorta above the iliac bifurcation. Compensatory blood flow to the placenta occurred through the adaptive increase of ovarian blood flow (Cunningham et al., 2018). The branches of the two ovarian arteries that supply the uterus were also ligated to prevent blood from flowing into the uterus through the ovarian artery.
The major changes were marked in red.
Question two:
Fig.1 should be positioned later in the text: in the present position is confusing, because it refers to Results.
Answer:
Thank you for your suggestion. I marked my revisions in red in the manuscript.
Question three:
Some sentences are awkward, i.e. "As shown in Fig. 2A, the increase in placenta of HIF1α levels that was observed in RUPP groups as compared with control groups (page 7); or "This study provided a new insight into the prevention and treatment of placenta-related diseases by phytochemicals." (page 3) written in the introduction, while it sounds as a results/discussion argument.
Answer:
Before revision:
As shown in Fig. 2A, the increase in placenta of HIF1α levels that was observed in RUPP groups as compared with control groups
This study provided a new insight into the prevention and treatment of placenta-related diseases by phytochemicals.
After revision:
HIF1α levels, LDH, and CS activities in the placental tissue were increased in the RUPP group in comparison to control group (Fig. 2A).
I marked my revisions in red in the manuscript.
Question four:
The legends of figures #1,2 and 3 are identical in the initial part. This is highly redundant and useless, as the procedures are explained in the Materials&Methods section.
Answer:
Thank you for your suggestion. The major changes were marked in red.
Question five:
The conclusions sounds a bit reckless. Placental insufficiency is an umbrella term which encompass many different conditions (immunological such as LES, or anatomical, such as uterine malformations, or related to placental implantation site, such as placenta previa), any of which can lead to abruptio placentae, IUGR, fetal demise, stillbirth etc. : reduced blood flow/pressure from spiral uterine arteries (mimicked by RUPP procedure) is not the only mechanism by which placental hypoxia takes place in humans. Maternal vascular malperfusion (the histologic pattern of placental hypoxia) may show also signs of acute atherosis, fibrinoid necrosis, chronic perivasculitis, trophoblasts persistence in placental implantation bed etc. The Authors should avoid to draw such relations in the regards of suggesting that the daily intake of citrus juice could help to prevent placental insufficiency in pregnant women. with such easiness, and leave the good results they achieved in the experimental area.
Answer:
Before revision:
The present study demonstrated that the supplementation of NOB during pregnancy in SD rats attenuated hypoxia-induced placental apoptosis and mitigated placental damage. No side effects were found in normal rats taking NOB during pregnancy. In vitro studies had found that the P53 signaling pathway was critical to alleviating the hypoxia-induced apoptosis of BeWo cells. Furthermore, the hypoxia-induced decrease of genomic DNA methylation and DNA methyltransferase levels in the placenta and BeWo cells were alleviated by NOB. Therefore, the research provided a new approach to preventing hypoxia-induced pregnancy-related diseases by incorporating NOB as a nutritional supplement.
After revision:
The present study demonstrated that NOB supplementation during pregnancy in-hibited placental apoptosis and attenuated placental damage in an SD rat model of RUPP. No side effects were found in normal rats taking NOB during pregnancy. In this process, NOB could restore genomic DNA methylation and DNA methyltransferase, and inhibit the P53 signaling pathway, resulting in the inhibition of the placental apoptosis and damage caused by RUPP. Therefore, the research provided a new approach to the prevention and clinical treatment of placenta-related diseases and the development and utilization of natural plant compounds.
The major changes were marked in red.
Question six:
Many references are outdated and should be substituted with more recent ones
Answer:
Thank you for your suggestion. The major changes were marked in red.
Question seven:
English revision is recommended.
Answer:
Thank you for your suggestion, and we have revised the manuscript.
I marked my revisions in red in the manuscript.
References
Cunningham, M. W., Jr., Castillo, J., Ibrahim, T., Cornelius, D. C., Campbell, N., Amaral, L., . . . LaMarca, B. (2018). AT1-AA (Angiotensin II Type 1 Receptor Agonistic Autoantibody) Blockade Prevents Preeclamptic Symptoms in Placental Ischemic Rats. Hypertension, 71(5), 886-893. https://doi.org/10.1161/HYPERTENSIONAHA.117.10681.
Williamson, R. D., McCarthy, F. P., Manna, S., Groarke, E., Kell, D. B., Kenny, L. C., & McCarthy, C. M. (2020). L-(+)-Ergothioneine Significantly Improves the Clinical Characteristics of Preeclampsia in the Reduced Uterine Perfusion Pressure Rat Model. Hypertension, 75(2), 561-568. https://doi.org/10.1161/HYPERTENSIONAHA.119.13929.
Reviewer 3 Report
In this study, the authors investigated the effect of the phytochemical (NOB) hypoxia induced placental insufficiency using an established animal model. The study is well-designed with an adequate number of individual rats for each of the treatment groups. Although not dramatic, NOB often had a statistically significant effect on multiple measures. To the authors’ credit, they do not overstate the significance of their findings. These data should serve as the foundation for follow up studies of the effect of this, and related compounds as a dietary intervention in the prevention/reversal of placental insufficiency in normal and high-risk individuals.
Specific comments
Although fine overall, there are occasional typos/grammatical errors that may be due to ESL
In the legend for figure 1, there is reference to the superscript ‘c’ indicating a significant difference relative to the hypoxia group, but I fail to see a ‘c’ anywhere on the figure. Was it left off accidentally?
Additional information regarding animal husbandry is needed. In particular, were the animals maintained under conventional or SPF conditions? Would the housing condition be expected to influence the outcomes? What was the nutrient formulation of the diet used? How long were the animals acclimated to the facility prior to the beginning of the experiment?
Were replicates included as a covariate in the statistical models? If not, was it known there was no significant difference(s) among the individual replicates?
In the discussion, the authors propose the consumption of a specific volume of citrus juice in light of the findings of this study, but their study utilized purified NOB. Inclusion of a citrus-fed treatment group would make these data more compelling if it paralleled what was seen with the purified compound. Have any citrus feeding studies been done? What were the results?
The authors seem to propose that the effect of NOB is p53-dependent, but it’s not entirely clear if this was their intent. Please rewrite the discussion of the putative role of p53.
Epigenetic programming is also invoked as a potential mechanism later in the discussion. Is this due to epigenetic effects on p53 expression? Is this the intent? Again, these last sections need to be rewritten so it is more clear.
Author Response
Responses to reviewer C comments
Question one:
Although fine overall, there are occasional typos/grammatical errors that may be due to ESL.
Answer:
Thank you for your suggestion, and we have revised the manuscript.
The major changes were marked in red.
Question two:
In the legend for figure 1, there is reference to the superscript ‘c’ indicating a significant difference relative to the hypoxia group, but I fail to see a ‘c’ anywhere on the figure. Was it left off accidentally?
Answer:
Thank you for your suggestion. Compared with the human gestation period of approximately 226 days, the rat gestation period is only 20-22 days, which is less than one tenth of the human gestation period, so the pregnant rat model has the characteristics of a short experimental period and economical affordability.
In this study, rats were given daily doses of 50 mg/kg body weight NOB or the same volume of 0.1% DMSO-saline solution from GD 0.5 to GD 17.5. On the GD 18.5, the rats in the treatment group underwent uterine artery pressure reduction operation, and the placenta and fetuses were collected 8 hours later to explore the protective effect of NOB during pregnancy on acute hypoxia in the late trimester of pregnancy. As shown in Fig.1, the liver and spleen coefficient of the female rats was lower in the RUPP and NOB+RUPP groups than the control groups. No difference was observed in weight and other organ coefficients (brain, heart, lung, and kidney) of female rats, litter size, fetal weight, and placental efficiency among all experimental groups.
The legend has been updated.
Question three:
Additional information regarding animal husbandry is needed. In particular, were the animals maintained under conventional or SPF conditions? Would the housing condition be expected to influence the outcomes? What was the nutrient formulation of the diet used? How long were the animals acclimated to the facility prior to the beginning of the experiment?
Answer:
Before revision:
Male and female SD rats (7-8 weeks old) were purchased from Weitong Lihua Laboratory Animal Technology Co., Ltd. (Beijing, China). The study was approved by the ethics committee of Capital Medical University (AEEI-2020-198). Rats were housed and maintained under standard laboratory conditions with controlled temperature at 20±2 oC and a 12:12-hour light/dark cycle. All rats were fed a standard rodent diet adlibitum, with free access to clean drinking water throughout the duration of the present study.
After revision:
All animal protocols and procedures used in this study were approved by the ethics committee of Capital Medical University (AEEI-2020-198). Specific pathogen-free (SPF) male and female Sprague-Dawley (SD) rats (7-8 weeks old) were purchased from Weitong Lihua Laboratory Animal Technology Co., Ltd. (Beijing, China). Rats were individually housed in conventional polypropylene cages (48 cm long, 33 cm wide, and 21 cm high) equipped with stainless steel food funnels and wood shavings as bedding. Cages were updated twice a week. The room was kept at 20 ± 2 oC, with 70% humidity, and a 12-hour light-dark cycle. All rats were given ad libitum access to a commercial rodent breeding diet (Keao xieli, Beijing, China) and water.
Table S1 Nutrient composition of rodent breeding feed
|
Component |
Content |
|
Moisture (%) |
8.00 |
|
Crude protein (%) |
22.30 |
|
Crude fat (%) |
4.70 |
|
Crude fiber (%) |
3.50 |
|
Coarse ash (%) |
6.60 |
|
Calcium (%) |
1.18 |
|
Phosphorus (%) |
0.78 |
|
Lysine (%) |
1.72 |
|
Methionine and cystine (%) |
0.78 |
|
Arginine (%) |
1.95 |
|
Histidine (%) |
0.80 |
|
Tryptophan (%) |
0.26 |
|
Phenylalanine and tyrosine (%) |
2.23 |
|
Threonine (%) |
1.04 |
|
Leucine (%) |
2.36 |
|
Isoleucine (%) |
1.23 |
|
Valine (%) |
1.46 |
|
Magnesium (%) |
0.28 |
|
Potassium (%) |
0.95 |
|
Natrium (%) |
0.33 |
|
Vitamin A (KIU/kg) |
19.00 |
|
Vitamin D (KIU/kg) |
2.50 |
|
Vitamin E (IU/kg) |
125.00 |
|
Vitamin K (mg/kg) |
10.00 |
|
Vitamin B1 (mg/kg) |
18.60 |
|
Vitamin B2 (mg/kg) |
18.60 |
|
Vitamin B6 (mg/kg) |
12.30 |
|
Niacin (mg/kg) |
119.00 |
|
Pantothenic acid (mg/kg) |
30.00 |
|
Folic acid (mg/kg) |
11.90 |
|
Biotin (mg/kg) |
0.40 |
|
Vitamin B12 (mg/kg) |
0.04 |
|
Choline (mg/kg) |
1900.00 |
|
Iron (mg/kg) |
250.00 |
|
Manganese (mg/kg) |
130.00 |
|
Copper (mg/kg) |
19.90 |
|
Zinc (mg/kg) |
67.00 |
|
Iodine (mg/kg) |
1.00 |
|
Selenium (mg/kg) |
0.20 |
The major changes were marked in red.
Question four:
Were replicates included as a covariate in the statistical models? If not, was it known there was no significant difference(s) among the individual replicates?
Answer:
After one week of adaptive feeding, virgin female rats were time-mated to male rats. Gestational day (GD) 0.5 was designated by the presence of a seminal plug or sperm in the vagina. All pregnant rats were healthy and in a good mood, and then randomly assigned to four treatment groups.
No covariates were used in the statistical model of this study. We averaged the independent repeated data showing no significant difference in the analysis of variance, otherwise, the experiment had to be repeated.
Question five:
In the discussion, the authors propose the consumption of a specific volume of citrus juice in light of the findings of this study, but their study utilized purified NOB. Inclusion of a citrus-fed treatment group would make these data more compelling if it paralleled what was seen with the purified compound. Have any citrus feeding studies been done? What were the results?
Answer:
Preliminary studies have found that NOB reduced the abnormal apoptosis of trophoblast cells of JEG-3 and BeWo caused by hypoxia (Zhang et al., 2021), and was safe for normal trophoblast cells (Zhang et al., 2020). In this study, the effects of NOB supplementation during pregnancy on hypoxia-induced placental damage in SD rats were further investigated, and it was found that NOB inhibited hypoxia-induced placental damage via modulating the P53 signaling pathway and global DNA methylation.
The literature about the effects of citrus fruits on pregnancy in the last 5 years was retrieved. A Japanese prenatal cohort study of risk and preventive factors for maternal and child health problems found that citrus fruit intake during pregnancy improved children's mood and behavior, as well as reduced the risk of hyperactivity (Miyake, Tanaka, Okubo, Sasaki, & Arakawa, 2020). A UK birth cohort study showed that citrus fruit intake reduced maternal nausea and vomiting in early pregnancy (Crozier, Inskip, Godfrey, Cooper, Robinson, & Group, 2017). A EXHES-Spain cohort study found that eating more citrus fruits reduced bisphenol A concentrations in the urine of pregnant women exposed to bisphenol A (Martinez et al., 2021). These evidences only suggest a beneficial effect of citrus fruit intake on pregnancy, but fail to clarify the intrinsic link between daily citrus fruit and pregnancy outcomes. Meanwhile, there is still no literature report on the intervention of citrus fruits on pregnant rats.
In the United States, NOB has been approved as a nutritional supplement, and is claimed to improve cognitive and metabolic function, promote cardiovascular function, and regulate circadian rhythm. However, in China, NOB has not yet been approved as healthy food or new food ingredients, which cannot be used by population and clinical research at present. Citrus peel is a food additive approved by the Food and Drug Administration in USA, and a drug homologous substance approved by National Health Commission in China. Therefore, in terms of future research and industrialization of NOB, citrus whole fruit juice or citrus peel can make up for the regulatory limitations of directly using NOB.
Your suggestion to study the effects of citrus whole fruit and juice feeding on pregnant animals is extremely meaningful and prospective. We will follow-up animal experiments to study the effects of citrus whole fruit and juice during pregnancy on hypoxia-induced placental damage in future.
Question six:
The authors seem to propose that the effect of NOB is p53-dependent, but it’s not entirely clear if this was their intent. Please rewrite the discussion of the putative role of p53.
Answer:
Before revision:
As an important tumor suppressor gene, P53 forms a tetrameric transcription factor that directly regulates about 500 target genes, thereby controlling cell processes including DNA breaks, apoptosis, and cell cycle caused by hypoxia [21]. NOB was bind to P53 protein by reducing free energy, and interacted with Leu145, Thr150, Pro151, Thr155, Val157, Cys220, Pro222, Pro223 and Thr230 in a flexible loop region of P53 protein by using molecular docking and molecular dynamics simulation [17]. The results of UV spectroscopy, fluorescence spectroscopy and circular dichroism further found that NOB could spontaneously bind to the P53 protein, enter the hydrophobic pocket of the P53 protein, cause the static quenching of the P53 protein, change the protein conformation, and finally affect the function and activity of the P53 protein [17]. In order to clarify the protective effect of P53 on hypoxic placental cell apoptosis, P53 gene was knockdown in BeWo cell lines for in vitro experiments. The results showed that NOB intervention alleviated hypoxia-induced apoptosis and cell cycle arrest in the G1 phase of BeWo cell lines, but in the cell line with P53 gene knockdown, NOB was less effective in alleviating apoptosis and cell cycle arrest. The above evidences suggested that the P53 signaling pathway was necessary for NOB’s anti-apoptosis effect.
After revision:
As an important tumor suppressor gene, P53 forms a tetrameric transcription factor that directly regulates about 500 target genes, thereby controlling cell processes including DNA breaks, apoptosis, and cell cycle caused by hypoxia (Boutelle & Attardi, 2021). Under mild hypoxia, P53 protein levels are decreased to protect cells from apoptosis and promote cell viability. While, under severe hypoxia, P53 protein levels are stabilized, thereby reducing HIF1α transcriptional activity and inducing apoptosis (Jing et al., 2019). A preliminary study found that NOB inhibited hypoxia-induced apoptosis of JEG-3 and BeWo trophoblast cells, and the severity of apoptosis was positively correlated with the level of P53 (Zhang et al., 2021). Molecular docking and molecular dynamics simulation found that NOB bonds to P53 protein by reducing free energy and interacted with Leu145, Thr150, Pro151, Thr155, Val157, Cys220, Pro222, Pro223 and Thr230 in a flexible loop region of P53 protein (Zhang et al., 2021). The results of UV spectroscopy, fluorescence spectroscopy, and circular dichroism further proved that NOB could spontaneously bind to the P53 protein, enter the hydrophobic pocket of the P53 protein, cause the static quenching of the P53 protein, change the protein conformation, and finally affect the function and activity of the P53 protein (Zhang et al., 2021). The above findings suggested that the P53 pathway may be one of the mechanisms by which NOB exerts anti-trophoblast apoptosis. To test this hypothesis, the in vitro experiments tested BeWo cell lines with the P53 gene knockdown. The results showed that NOB intervention alleviated hypoxia-induced apoptosis and cell cycle arrest in the G1 phase of BeWo cell lines, but in the cell line with P53 gene knockdown, NOB was less effective in alleviating apoptosis and cell cycle arrest. Those findings indicated that the P53 signaling pathway was critical for NOB’s anti-apoptosis effect.
The major changes were marked in red.
Question seven:
Epigenetic programming is also invoked as a potential mechanism later in the discussion. Is this due to epigenetic effects on p53 expression? Is this the intent? Again, these last sections need to be rewritten so it is more clear.
Answer:
Before revision:
Epigenetic programming during pregnancy is extremely sensitive to environmental impact [37]. During this period, exposure to unfavorable environmental factors will significantly change the degree of placental DNA methylation and cause placental defects, leading to disrupted fetal development and adverse pregnancy outcomes [38]. In general, total genomic DNA hypomethylation highly correlates with adverse outcomes. A MARBLES cohort study found a lower degree of placental DNA methylation in pesticide-exposed mothers [39]. Genome-wide DNA methylation of mice’s placenta was reduced by exposure to bisphenol A during pregnancy, causing poor placental development and fetal health [40]. This study found that hypoxia reduced the degree of genome-wide DNA methylation in rat placental cells and BeWo cells, and compromised the levels of DNA methyltransferase, which were alleviated after NOB intervention.
After revision:
Epigenetic programming during pregnancy is extremely sensitive to environmental impacts (Tian et al., 2020). During this period, exposure to unfavorable environmental factors will significantly change the degree of placental DNA methylation and cause placental defects, leading to disrupted fetal development and adverse pregnancy outcomes (Pacini et al., 2020). In general, total genomic DNA hypomethylation highly correlates with adverse outcomes. A French mother-infant cohort study found a lower degree of placental DNA methylation in mothers exposed to phthalates (Jedynak et al., 2022). Genome-wide DNA methylation of mice’s placenta was reduced by the exposure to polychlorinated biphenyls during pregnancy, causing fetal neurodevelopmental disorders (Laufer et al., 2022). This study found that hypoxia reduced the degree of genome-wide DNA methylation in rat placental cells and BeWo cells, and decreased the levels of DNA methyltransferase, which were alleviated after NOB intervention. Temozolomide, as a DNA methylation activator, promoted apoptosis in glioblastoma cells by activating P53 and was dependent on sustained P21 induction (Aasland et al., 2019). DNA hypomethylating drugs, including 5-azacytidine and decitabine, induced apoptosis of colon cancer cells in a p53-dependent manner (Tong et al., 2020). All-trans retinoic acid downregulated the protein and enzymatic activity of DNA methyltransferases DNMT1, DNMT3a, and DNMT3b, thereby activating P53-dependent apoptosis of human hepatocytes (Choi, Jeong, & Jang, 2017). In vitro and in vivo nasopharyngeal carcinoma models found that the down-regulation of DNMT3B significantly caused the G1 phase arrest and promoted apoptosis by restoring P53 and P21 signaling pathways, thereby enhancing the radiosensitivity of nasopharyngeal carcinoma (Wu et al., 2020). These proofs suggest that NOB might inhibit the P53 signaling pathway by increasing the levels of genome-wide DNA methylation and DNA methyltransferase, finally alleviating the induced placental cell apoptosis.
The major changes were marked in red.
References
Aasland, D., Gotzinger, L., Hauck, L., Berte, N., Meyer, J., Effenberger, M., . . . Christmann, M. (2019). Temozolomide Induces Senescence and Repression of DNA Repair Pathways in Glioblastoma Cells via Activation of ATR-CHK1, p21, and NF-kappaB. Cancer Res, 79(1), 99-113. https://doi.org/10.1158/0008-5472.CAN-18-1733.
Boutelle, A. M., & Attardi, L. D. (2021). p53 and Tumor Suppression: It Takes a Network. Trends Cell Biol, 31(4), 298-310. https://doi.org/10.1016/j.tcb.2020.12.011.
Choi, J. H., Jeong, H., & Jang, K. L. (2017). Hepatitis B virus X protein suppresses all-trans retinoic acid-induced apoptosis in human hepatocytes by repressing p14 expression via DNA methylation. J Gen Virol, 98(11), 2786-2798. https://doi.org/10.1099/jgv.0.000958.
Crozier, S. R., Inskip, H. M., Godfrey, K. M., Cooper, C., Robinson, S. M., & Group, S. W. S. S. (2017). Nausea and vomiting in early pregnancy: Effects on food intake and diet quality. Matern Child Nutr, 13(4). https://doi.org/10.1111/mcn.12389.
Cunningham, M. W., Jr., Castillo, J., Ibrahim, T., Cornelius, D. C., Campbell, N., Amaral, L., . . . LaMarca, B. (2018). AT1-AA (Angiotensin II Type 1 Receptor Agonistic Autoantibody) Blockade Prevents Preeclamptic Symptoms in Placental Ischemic Rats. Hypertension, 71(5), 886-893. https://doi.org/10.1161/HYPERTENSIONAHA.117.10681.
Jedynak, P., Tost, J., Calafat, A. M., Bourova-Flin, E., Broseus, L., Busato, F., . . . Philippat, C. (2022). Pregnancy exposure to phthalates and DNA methylation in male placenta - An epigenome-wide association study. Environ Int, 160, 107054. https://doi.org/10.1016/j.envint.2021.107054.
Jing, X., Yang, F., Shao, C., Wei, K., Xie, M., Shen, H., & Shu, Y. (2019). Role of hypoxia in cancer therapy by regulating the tumor microenvironment. Mol Cancer, 18(1), 157. https://doi.org/10.1186/s12943-019-1089-9.
Laufer, B. I., Neier, K., Valenzuela, A. E., Yasui, D. H., Schmidt, R. J., Lein, P. J., & LaSalle, J. M. (2022). Placenta and fetal brain share a neurodevelopmental disorder DNA methylation profile in a mouse model of prenatal PCB exposure. Cell Rep, 38(9), 110442. https://doi.org/10.1016/j.celrep.2022.110442.
Martinez, M. A., Gonzalez, N., Marti, A., Marques, M., Rovira, J., Kumar, V., & Nadal, M. (2021). Human biomonitoring of bisphenol A along pregnancy: An exposure reconstruction of the EXHES-Spain cohort. Environ Res, 196, 110941. https://doi.org/10.1016/j.envres.2021.110941.
Miyake, Y., Tanaka, K., Okubo, H., Sasaki, S., & Arakawa, M. (2020). Maternal consumption of vegetables, fruit, and antioxidants during pregnancy and risk for childhood behavioral problems. Nutrition, 69, 110572. https://doi.org/10.1016/j.nut.2019.110572.
Pacini, G., Paolino, S., Andreoli, L., Tincani, A., Gerosa, M., Caporali, R., . . . Cutolo, M. (2020). Epigenetics, pregnancy and autoimmune rheumatic diseases. Autoimmun Rev, 19(12), 102685. https://doi.org/10.1016/j.autrev.2020.102685.
Tian, F. Y., Everson, T. M., Lester, B., Punshon, T., Jackson, B. P., Hao, K., . . . Marsit, C. J. (2020). Selenium-associated DNA methylation modifications in placenta and neurobehavioral development of newborns: An epigenome-wide study of two U.S. birth cohorts. Environ Int, 137, 105508. https://doi.org/10.1016/j.envint.2020.105508.
Tong, R., Wu, X., Liu, Y., Liu, Y., Zhou, J., Jiang, X., . . . Ma, L. (2020). Curcumin-Induced DNA Demethylation in Human Gastric Cancer Cells Is Mediated by the DNA-Damage Response Pathway. Oxid Med Cell Longev, 2020, 2543504. https://doi.org/10.1155/2020/2543504.
Williamson, R. D., McCarthy, F. P., Manna, S., Groarke, E., Kell, D. B., Kenny, L. C., & McCarthy, C. M. (2020). L-(+)-Ergothioneine Significantly Improves the Clinical Characteristics of Preeclampsia in the Reduced Uterine Perfusion Pressure Rat Model. Hypertension, 75(2), 561-568. https://doi.org/10.1161/HYPERTENSIONAHA.119.13929.
Wu, C., Guo, E., Ming, J., Sun, W., Nie, X., Sun, L., . . . Hu, G. (2020). Radiation-Induced DNMT3B Promotes Radioresistance in Nasopharyngeal Carcinoma through Methylation of p53 and p21. Mol Ther Oncolytics, 17, 306-319. https://doi.org/10.1016/j.omto.2020.04.007.
Zhang, M., Liu, J., Zhang, R., Liang, Z., Ding, S., Yu, H., & Shan, Y. (2021). Nobiletin, a hexamethoxyflavonoid from citrus pomace, attenuates G1 cell cycle arrest and apoptosis in hypoxia-induced human trophoblast cells of JEG-3 and BeWo via regulating the p53 signaling pathway. Food Nutr Res, 65. https://doi.org/10.29219/fnr.v65.5649.
Zhang, M., Zhang, R., Liu, J., Wang, H., Wang, Z., Liu, J., . . . Yu, H. (2020). The Effects of 5,6,7,8,3',4'-Hexamethoxyflavone on Apoptosis of Cultured Human Choriocarcinoma Trophoblast Cells. Molecules, 25(4). https://doi.org/10.3390/molecules25040946.
Round 2
Reviewer 2 Report
The Authors extensively revised the work, which acquired more scientific soundness and it is now more coherent through the various sections.
In the Conclusions, were the Authors stated "the research provided a new approach to the prevention and clinical treatment of placenta-related diseases" it would be preferable to state that "the research provided a new approach to the prevention and clinical treatment of some placental hypoxia-related diseases".
Still some minor spell check required.
Author Response
Question one:
In the Conclusions, were the Authors stated "the research provided a new approach to the prevention and clinical treatment of placenta-related diseases" it would be preferable to state that "the research provided a new approach to the prevention and clinical treatment of some placental hypoxia-related diseases".
Answer:
Thank you for your suggestion. I marked my revisions in red in the manuscript.
Question two:
Still some minor spell check required.
Answer:
Thank you for your suggestion. I marked my revisions in red in the manuscript.
